# Pathophysiology and Treatment of Diabetic Cardiomyopathy and Heart Failure in Patients with Diabetes Mellitus

**DOI:** 10.3390/ijms23073587

**Published:** 2022-03-25

**Authors:** Kazufumi Nakamura, Toru Miyoshi, Masashi Yoshida, Satoshi Akagi, Yukihiro Saito, Kentaro Ejiri, Naoaki Matsuo, Keishi Ichikawa, Keiichiro Iwasaki, Takanori Naito, Yusuke Namba, Masatoki Yoshida, Hiroki Sugiyama, Hiroshi Ito

**Affiliations:** 1Department of Cardiovascular Medicine, Faculty of Medicine, Dentistry and Pharmaceutical Sciences, Okayama University, Okayama 700-8558, Japan; miyoshit@cc.okayama-u.ac.jp (T.M.); yoshid-m@cc.okayama-u.ac.jp (M.Y.); akagi-s@cc.okayama-u.ac.jp (S.A.); p5438a3l@okayama-u.ac.jp (Y.S.); eziken82@gmail.com (K.E.); phff2e2u@s.okayama-u.ac.jp (N.M.); ichikawa1987@gmail.com (K.I.); puk75d68@okayama-u.ac.jp (K.I.); p1o40ji8@s.okayama-u.ac.jp (T.N.); pvnv8vve@s.okayama-u.ac.jp (Y.N.); pb0v2jyw@okayama-u.ac.jp (M.Y.); itomd@md.okayama-u.ac.jp (H.I.); 2Department of Internal Medicine, Okayama Saiseikai General Hospital, Okayama 700-8511, Japan; circ.sugiyamah@gmail.com

**Keywords:** heart failure, lipotoxicity, SGLT2 inhibitor

## Abstract

There is a close relationship between diabetes mellitus and heart failure, and diabetes is an independent risk factor for heart failure. Diabetes and heart failure are linked by not only the complication of ischemic heart disease, but also by metabolic disorders such as glucose toxicity and lipotoxicity based on insulin resistance. Cardiac dysfunction in the absence of coronary artery disease, hypertension, and valvular disease is called diabetic cardiomyopathy. Diabetes-induced hyperglycemia and hyperinsulinemia lead to capillary damage, myocardial fibrosis, and myocardial hypertrophy with mitochondrial dysfunction. Lipotoxicity with extensive fat deposits or lipid droplets is observed on cardiomyocytes. Furthermore, increased oxidative stress and inflammation cause cardiac fibrosis and hypertrophy. Treatment with a sodium glucose cotransporter 2 (SGLT2) inhibitor is currently one of the most effective treatments for heart failure associated with diabetes. However, an effective treatment for lipotoxicity of the myocardium has not yet been established, and the establishment of an effective treatment is needed in the future. This review provides an overview of heart failure in diabetic patients for the clinical practice of clinicians.

## 1. Complication of Diabetes and Heart Failure

Heart failure is one of the most common complications of diabetes mellitus, and diabetes mellitus is an independent risk factor for heart failure. The Framingham study showed that the development of heart failure is two times more likely in men with diabetes and five times more likely in women with diabetes than in non-diabetics [1]. On the other hand, the complication rate of diabetes in patients with heart failure is 30–40% [2], supporting the close relationship between diabetes and heart failure.

The basis of the link between diabetes and heart failure is not only the complication of ischemic heart disease, but also metabolic disorders such as glucose toxicity and lipotoxicity based on insulin resistance, vascular endothelial dysfunction, microcirculatory disorder, and capillary failure. Various mechanisms are involved. It has been proposed that cardiac dysfunction in the absence of significant coronary artery disease, hypertension, and valvular heart disease should be called “diabetic cardiomyopathy” [3,4,5,6]. However, its existence has been debated for many years [4,7,8]. Diabetic cardiomyopathy is described in the American College of Cardiology (ACC)/American Heart Association (AHA) Heart Failure Guidelines published in 2013 [9]. On the other hand, in the recent guidelines of the European Society of Cardiology (ESC) and the European Society of Diabetes (EASD) on diabetes and cardiovascular disease, it is stated that diabetic cardiomyopathy has not been established as a unique clinical entity, and future development is awaited [2]. Data on diabetic cardiomyopathy reported so far are presented in this review. This review provides an overview of heart failure in diabetic patients for the clinical practice of clinicians.

Patients with obesity, insulin resistance, and dyslipidemia show similar cardiac dysfunction, and in the absence of diabetes, the condition may be called lipotoxic cardiomyopathy or obesity-related cardiomyopathy [10].

## 2. History of Diabetic Cardiomyopathy

Then, in 1972, Rubler et al. reported that there was no evidence of coronary artery disease in autopsy results for four patients with diabetic glomerulosclerosis and heart failure [11]. Myocardial hypertrophy and fibrosis were noted in the hearts of those patients, suggesting that the metabolism is responsible for this result. His description meets the contemporary definition of cardiomyopathy of the European Society of Cardiology [12]. Rubler’s observations were supported by results reported by Regan in 1977 [13]. Autopsies of 11 uncomplicated diabetic patients revealed that 9 had no coronary artery disease and most of them died of heart failure. Collagen accumulation was present as perivascular, intermuscular, or replacement fibrosis. Multiple samples of the left ventricle and septum revealed increased levels of triglycerides and cholesterol compared to controls. Therefore, it was suggested that a diffuse extravascular abnormality is a basis for cardiomyopathic features in diabetes.

## 3. Pathophysiology and Mechanism of Heart Failure Associated with Diabetes

There are different types of heart failure associated with diabetes. Except for coronary artery disease, the frequency of left ventricular diastolic dysfunction is as high as 40–60% [14]. It has been reported that approximately 40% of patients with heart failure with preserved ejection fraction (HFpEF) have diabetes, and diabetes is thought to be closely related to the pathophysiology of HFpEF [15]. Left ventricular diastolic dysfunction is the initial symptom, and left ventricular systolic dysfunction and heart failure with reduced ejection fraction (HFrEF) often appear as diabetes progresses.

Table 1 shows the currently considered mechanism of heart failure associated with diabetes, and it is not limited to diabetic cardiomyopathy. Diabetes-induced hyperglycemia and hyperinsulinemia lead to capillary damage, myocardial fibrosis, and myocardial hypertrophy with mitochondrial dysfunction [5,16,17,18,19,20,21,22,23]. Lipotoxicity with extensive fat deposits or lipid droplets is observed on cardiomyocytes [23,24,25,26]. Furthermore, increased oxidative stress and inflammation cause cardiac fibrosis and hypertrophy.

## 4. Lipotoxicity to the Myocardium

Excessive circulating free fatty acids that are increased due to diabetes and obesity accumulate in the adipose tissue mainly as triglycerides. Ectopic fat that accumulates in organs other than the adipocytes of visceral fat and subcutaneous fat causes the dysfunction of cells and organs, such as the liver, pancreatic β cells, the skeletal muscle, and myocardium, through the deterioration of mitochondrial function [25]. This is called lipotoxicity. Chronic inflammation and insulin resistance occur due to fat accumulation in the liver and skeletal muscle, resulting in impaired glucose tolerance, dyslipidemia, and hypertension.

In the heart, fat accumulation is present around the heart and in the myocardium. Pericardial fat is divided into two types, paracardial fat located on the outside and epicardial fat located on the inside, based on the epicardium. High epicardial fat mass measured by computed tomography in a heart disease-free population has been reported to be an independent predictor of the development of coronary artery disease [26].

Figure 1 shows the mechanism of lipotoxicity to the myocardium itself. An increase in free fatty acids in the blood leads to an increase in fatty acids in cardiomyocytes. Excess fatty acids accumulate in cells as lipid droplets and triglycerides, and, at the same time, diacylglycerol and ceramide, which is one of the sphingolipids, also increase [10]. Diacylglycerol causes the exacerbation of insulin resistance and oxidative stress through the activation of protein kinase C (PKC). The level of diacylglycerol is increased, accompanied by increased membrane localization of PKC and decreased Akt activity in human failing myocardium [27]. These observations suggest that diacylglycerol is a toxic lipid intermediate in the heart. Ceramide causes mitochondrial dysfunction and oxidative stress. C6-ceramide reduces Akt activity and increases brain natriuretic peptide (*BNP*) mRNA expression in cardiomyocytes, and the inhibition of ceramide biosynthesis in the heart improves lipotoxic cardiomyopathy [28]. These results suggest that ceramide accumulation contributes to the development of left ventricular hypertrophy and cardiac dysfunction. It has been reported that this lipotoxicity mainly impairs diastolic function [29].

In diabetic cardiomyopathy, excess fatty acids activate peroxisome proliferator-activated receptor α (PPARα), further increase fatty acid uptake via CD36, and promote lipotoxicity. However, it has been reported that the activity of PPARα is reduced in hypertrophic and failing hearts that are not caused by diabetes, resulting in β-oxidation attenuation of fatty acids and energy deficiency [30]. Since there is no consensus on heart failure in general, further studies are needed to clarify this point.

## 5. Increased Oxidative Stress in Myocardium

The mechanisms by which diabetes increases oxidative stress are as follows: (1) oxidative stress due to the abnormal regulation of the electron transport chain in mitochondria, (2) enhanced renin-angiotensin system (RAS) and nicotinamide adenine dinucleotide phosphate (NADPH) oxidase activity, and (3) increased oxidative stress due to advanced glycation end products (AGEs) [5]. AGEs not only directly damage cells but also increase oxidative stress by increasing the production of reactive oxygen species (ROS) [5]. Secretory extracellular AGEs activate NADPH oxidase and increase oxidative stress and inflammation by binding to cell surface receptors for advanced glycation end products (RAGE). Oxidative stress causes cardiac fibrosis and hypertrophy [5,31,32].

High glucose has been shown to inhibit Nrf2- and Sirt1-mediated antioxidant signaling and activate NF-κB-mediated inflammatory signaling [33]. Oxidative stress and inflammation interact to increase the production of ROS and inflammatory factors, which promote and exacerbate cardiac dysfunction and remodeling.

Hydrogen sulfide (H2S), as the third gasotransmitter, may play an important role in the cardiovascular system. H2S deficiency aggravated mitochondrial damage, increased ROS accumulation, promoted necroptosis, activated NLRP3 inflammasome, and finally exacerbated diabetic cardiomyopathy in the hearts of mice [34].

## 6. Mitochondrial Dysfunction

Insulin resistance leads to lower glucose utilization and oxidative reduction, which results in an imbalance in the uptake and oxidation of fatty acids. Finally, they lead to mitochondrial dysfunction [35]. Furthermore, in diabetes, progressive mitochondrial impairment in cardiomyocytes causes lipid accumulation and results in the generation of a large amount of ROS, which increase oxidative stress, worsening the diabetic cardiomyopathy and further impairing myocardial function [36].

Mitophagy is a type of autophagy that occurs in dysfunctional mitochondria, and it plays a key role in mitochondrial quality control. Mitophagy plays a protective role in diabetic cardiomyopathy, principally through the clearance of abnormal mitochondria, which prevents oxidative stress and reduces myocardial apoptosis [35]. However, excessive mitophagy may exacerbate myocardial damage in patients with diabetic cardiomyopathy [35]. Several signaling pathways that regulate mitophagy, including PINK1/parkin, AMPK-mTOR, and Wnt pathways, have been identified.

## 7. Inflammation

Diabetes causes a chronic inflammatory condition mediated by increased inflammasome. The nucleotide-binding oligomerization domain-like receptor family, pyrin domain-containing 3 (NLRP3) inflammasome is associated with the development of diabetic cardiomyopathy [37,38]. High FFA levels, impaired insulin metabolic signaling, and hyperglycemia activate NLRP3. Activated NLRP3 induces interleukin-1 beta (IL-1β) and interleukin-18 (IL-18) production and causes local tissue inflammation. Nuclear factor-kB (NF-kB) and thioredoxin-interacting/-inhibiting protein (TXNIP) mediate the ROS-induced caspase-1 and IL-1beta activation, which are the effectors of NLRP3 inflammasome [37]. NLRP3 inflammasome–caspase-1-mediated pyroptosis, a necrotic form of regulated cell death [39], was found in the myocardium of diabetic rats [37]. *NLRP3* gene silencing therapy ameliorated cardiac inflammation, pyroptosis, fibrosis, and cardiac function [37].

## 8. Abnormal Myocardial Calcium Handling

Type II diabetes in mice leads to impaired contractility and relaxation due to elevated intracellular resting Ca^2+^, slowed Ca^2+^ transients, the reduction of sarcoplasmic reticulum Ca^2+^ pumping, and the impairment of sarcoplasmic reticulum Ca^2+^ reuptake [40,41].

## 9. Autonomic Dysregulation in the Heart

Cardiac autonomic dysregulation or neuropathy includes abnormalities in heart rate control, vascular hemodynamics, and cardiac structure and function [42]. Hyperglycemia is a primary culprit, inducing oxidative stress and toxic glycosylation products, which results in neuronal dysfunction and death.

## 10. Other Mechanisms

Other pathogenic mechanisms of heart failure associated with diabetes include sodium retention due to hyperinsulinemia and vascular endothelial dysfunction. The vascular endothelium regulates the contraction/relaxation of the vascular wall, the adhesion of inflammatory cells to the vascular wall, vascular permeability, and the coagulation/fibrinolytic system. Among them, nitric oxide (NO) synthesized by endothelial nitric oxide synthase (eNOS) shows vascular smooth muscle relaxation and is important for vascular tone control.

Endothelial dysfunction is observed in patients with heart failure [43] and it is associated with an increased mortality risk in subjects with both ischemic and nonischemic heart failure [44]. In type 2 diabetes, vascular endothelial dysfunction is seen from the early stage of impaired glucose tolerance and is deeply involved in the development and progression of arteriosclerosis and coronary artery disease. Factors that cause vascular endothelial dysfunction include hyperglycemia, hyperinsulinemia associated with insulin resistance, hypoglycemia, and non-fasting hyperlipidemia (postprandial hyperlipidemia) [45,46]. In hyperglycemic conditions, glucose uptake into endothelial cells via glucose transporter 1 (GLUT 1) is increased. After that, protein kinase C (PKC) signaling and the formation of AGEs are enhanced, resulting in intracellular metabolic disorders and endothelial cell dysfunction. The phosphoinositide 3-kinase/Akt (PI3K/Akt) pathway, which is one of the insulin signaling pathways, has NO-producing and vasodilatory effects on vascular endothelial cells. Insulin resistance causes impaired signaling in this pathway, leading to progressive vascular endothelial dysfunction [47,48]. Hypoglycemia is thought to cause vascular endothelial dysfunction due to an increase in reactive oxygen species and catecholamines [49] and an increase in inflammatory cytokines [50]. Rather than persistent hyperglycemia, a steep rise in blood glucose due to postprandial hyperglycemia, that is, glycemic fluctuation, increases the production of inflammatory cytokines and oxidative stress and impairs vascular endothelial function. In fact, vascular endothelial dysfunction, as assessed by flow-mediated dilatation (FMD), was shown to be more severe in a group of patients with glycemic fluctuation than in a group of patients with persistent hyperglycemia.

Blood triglyceride (TG)-rich lipoprotein is increased through the dietary effects of increased chylomicron remnants (extrinsic) and increased VLDL synthesis in the liver (intrinsic). Furthermore, the delayed clearance of TG-rich lipoprotein causes non-fasting hypertriglyceridemia (non-fasting hyperlipidemia). In particular, obese patients show hypertriglyceridemia after meals from an early stage, and those with insulin resistance (metabolic syndrome patients) have prolonged high TG levels. This non-fasting hyperlipidemia impairs vascular endothelial function [45]. It is thought that the increased chylomicrons and VLDL remnants induce the production of inflammatory cytokines and oxidative stress and further attenuate the activity of eNOS and impair vascular endothelial function [45,51]. When a high-fat diet is consumed, blood TG, ApoB-48, and remnant-like lipoprotein cholesterol levels increase and peak at 4 h after eating. Interestingly, postprandial brachial artery FMD decreases and reaches the lowest level at the 4th hour after oral fat load, and decreased vascular endothelial function was observed [45].

## 11. What Is an Effective Treatment?

Treatment of HF is similar for patients with and without diabetes. However, antidiabetic drugs have different effects in HF patients, and it is necessary to prioritize drugs that are safe and reduce HF-related events [52]. We describe the effects of several drugs in this review. However, diabetic cardiomyopathy is a specific entity, while the drugs described below may or may not have effects on different cardiovascular endpoints in subjects with diabetes.

Since there is a U-curve phenomenon between HbA1c and mortality in diabetic patients with heart failure [53], it is difficult to improve the prognosis simply by lowering blood glucose. In addition, it is assumed that hypoglycemia has adverse effects on the cardiovascular system through the activation of sympathetic nerves and inflammation, and it is, therefore, important to avoid the occurrence of hypoglycemia.

There are several negative opinions about the effectiveness of intensive glycemic control in the primary prevention of heart failure. In a meta-analysis in which intensive glycemic control was compared with conventional glycemic control, it was shown that intensive therapy did not reduce the risk of hospitalization due to heart failure [54]. It has also been pointed out that intensive glucose-lowering treatment may exacerbate heart failure. A meta-analysis of 13 studies in 34,533 T2DM patients showed that intensive glucose-lowering treatment did not reduce cardiovascular events, but instead increased the risk of developing heart failure by 47% [55]. One of the mechanisms was the involvement of hypoglycemia, which is increased by intensive glucose-lowering treatment and the accompanying activation of the sympathetic nervous system.

### 11.1. Calorie Restriction

The basis of diabetes treatment is diet and exercise therapy, and if hypertension or heart failure is complicated, salt intake should be further reduced. Regarding calories, it is recommended to take calories based on the formula of “appropriate daily energy amount (kcal) = standard weight (kg) × physical activity amount”. In short-lived model organisms, such as yeast, nematodes, flies, and mice, calorie restriction extends their lifespan. Studies in healthy humans have also shown improved health-related quality of life and reduced oxidative stress in a 2-year 15% calorie-restricted group compared to those in a group without restriction [56]. Whether or not calorie restriction will be an effective treatment for human heart failure remains to be seen.

### 11.2. Sulfonylureas

Sulfonylureas may increase the risk of developing heart failure. In a retrospective study, the risk of myocardial infarction, total mortality, and the risk of developing heart failure were compared among groups of diabetes treatment drugs [57]. It has been reported that total mortality increased by 24–61% and heart failure increased by 18–30% when sulfonylureas alone were used compared to metformin monotherapy.

### 11.3. Insulin

Insulin is needed in patients with type 1 diabetes and for the control of hyperglycemia in some patients with type 2 diabetes. Since sodium retention is enhanced in the kidney, there is a concern that fluid retention in patients with heart failure may be exacerbated.

### 11.4. Thiazolidinediones

Pioglitazone, a thiazolidinedione, increases the risk of developing heart failure [58] and is not used in patients with heart failure. However, pioglitazone does not reduce cardiac function. Pioglitazone promotes Na + reabsorption and promotes fluid retention by activating sodium transporters in the proximal tubule and epithelial sodium channels in the collecting duct via peroxisome proliferator-activated receptor γ (PPARγ). On the other hand, pioglitazone is often used for secondary preventive purposes because it has a strong preventive effect on cardiovascular events [57]. In that case, to prevent the onset of heart failure due to fluid retention, it is recommended to use it in combination with a mineralocorticoid receptor antagonist or thiazide diuretics.

### 11.5. Dipeptidyl Peptidase-4 (DPP4) Inhibitors

There is little evidence that DPP4 inhibitors have a preventive effect on heart failure. It has been reported that DPP-4 inhibitors do not improve left ventricular diastolic function in patients with diabetes [59]. DPP4 inhibitors did not show a significant increase in the risk of developing heart failure compared to the placebo group in many long-term cardiovascular safety studies [60,61,62]. However, in the SAVOR-TIMI 53 study, it was shown that there was a risk of developing heart failure in the saxagliptin group [63]. There was no difference between the saxagliptin and placebo groups in rates of cardiovascular death, myocardial infarction, and stroke in patients with a history of cardiovascular events or in high-risk patients. However, in the saxagliptin group, hospitalization for heart failure was significantly increased compared to that in the placebo group. The reason is that DPP4 degrades not only glucagon-like peptide-1 (GLP-1), but also bioactive proteins such as substance P and neuropeptide Y [64]. DPP4 inhibitors increase sympathetic nerve activity by increasing their bioactive proteins, which may be involved in the onset of heart failure. Although no conclusions have been reached as to whether DPP4 inhibitors increase the risk of developing heart failure, DPP4 inhibitors are unlikely to reduce the risk of developing heart failure in diabetics.

### 11.6. Metformin

In the recent guidelines of the ESC for the diagnosis and treatment of acute and chronic heart failure [52], it is stated that metformin is thought to be safe in patients with HF, compared with insulin and sulfonylureas, on the basis of the results of observational studies [65,66]. Metformin is not recommended in patients with an eGFR < 30 mL/min/1.73 m^2^ because of the risk of lactic acidosis. So far, there has been no randomized controlled trial [52]. One of the mechanisms of action of metformin is the activation of AMP-activated protein kinase (AMPK). AMPK is an enzyme that regulates energy metabolism in various tissues including the heart, liver, and muscles. Metformin improved left ventricular function and prolonged the survival of mice with heart failure due to ischemia [67]. Metformin improved cardiac function in a dog model of pacing-induced heart failure [68]. The mechanism was mediated by AMP-kinase activation. These results suggest the existence of direct myocardial protection and heart failure-improving effect of metformin that does not depend on blood glucose improvement.

### 11.7. Sodium Glucose Cotransporter 2 (SGLT2) Inhibitor

In the EMPA-REG OUTCOME trial, empagliflozin, an SGLT2 inhibitor, suppressed heart failure hospitalization and heart failure death in patients with type 2 diabetes with a history of cardiovascular disease [69]. Similarly, in the CANVAS trial, canagliflozin, an SGLT2 inhibitor, reduced hospitalization for heart failure in patients with type 2 diabetes and a high cardiovascular risk [70]. Given these findings, SGLT2 inhibitors seem to be effective in preventing heart failure in patients with type 2 diabetes who have a history of cardiovascular disease, but primary prevention and effectiveness in the elderly aged 75 years or over are considered to be issues for the future [71]. The recently reported DECLARE-TIMI58 study included many cases of primary prevention of cardiovascular disease (59%) [72]. Dapagliflozin, an SGLT2 inhibitor, significantly suppressed hospitalization for heart failure in patients with or without a history of heart failure. It is expected that SGLT2 inhibitors will have a wide range of preventive effects on heart failure in diabetic patients in the future.

Many mechanisms are considered to be the mechanism by which SGLT2 inhibitors prevent and improve heart failure (Figure 2) [73,74,75,76].

There are two major mechanisms by which SGLT2 inhibitors prevent and improve heart failure, including metabolic and hemodynamic mechanisms. Metabolic mechanisms include a hypoglycemic effect, protection from lipotoxicity, weight loss, an increase in ketone in blood, a decrease in insulin, and the improvement of insulin resistance. Hemodynamic mechanisms include a diuretic effect and a decrease in blood pressure. As for protection from lipotoxicity, SGLT2 inhibitors decrease lipid accumulation in visceral fat and adipose lipolysis. The protective effect against lipotoxicity to the myocardium is not yet clear. Further studies are needed to clarify this point. Regarding the blood pressure-lowering effect, a blood pressure-lowering effect of about −3 mmHg was also observed in the above-mentioned EMPA-REG OUTCOME, CANVAS test, and DECLARE-TIMI58 trials. We also investigated the effect of tofogliflozin, an SGLT2 inhibitor, in dahl salt-sensitive rats fed a high-salt and high-fat diet. A blood pressure-lowering effect of about −3 mmHg and a cardiac hypertrophy-suppressing effect were observed [77].

A decrease in sympathetic nerve activity by SGLT2 inhibitors has been proposed as an additional mechanism [74]. A common feature of type 2 diabetes mellitus (T2DM) is the chronic activation of the sympathetic nervous system. SGLT2 inhibitors improved the circadian rhythm of sympathetic activity in rats with metabolic syndrome [78] and reduced the high-fat-diet-induced elevation of tyrosine hydroxylase and noradrenaline in the kidneys and hearts of mice [79]. In patients with T2DM, a higher heart rate (HR) is associated with an increased risk of death and cardiovascular complications. Among patients with a resting HR > 70 beats per minute, a higher HR was associated with a larger decrease after starting treatment with luseogliflozin, an SGLT2 inhibitor [80].

Recently, it has been reported that empagliflozin attenuates the late component of cardiac sodium channel current (late-*I*_Na_) [81]. The induction of late-*I*_Na_ is involved in the etiology of heart failure and arrhythmias [82], and drugs that are known to inhibit late-INa, such as ranolazine, reduce diastolic calcium loading in heart failure and long QT syndrome 3 (LQT3) models [83,84]. It was shown by using molecular docking techniques that empagliflozin binds to Nav1.5 in the same region as local anesthetics and ranolazine. The cardiac sodium channel may, therefore, be an important molecular target in the heart for SGLT2 inhibitors that significantly contributes to their beneficial effects against heart failure.

In patients with HFrEF and with or without type 2 diabetes, dapagliflozin in addition to standard treatment for heart failure significantly reduced the primary composite endpoint of worsening heart failure (hospitalization or an urgent visit resulting in intravenous therapy for heart failure) or cardiovascular death compared to those in the placebo group (DAPA-HF) [85]. Empagliflozin significantly reduced the primary composite endpoint of cardiovascular death or hospitalization for worsening heart failure compared to that in the placebo group in patients with HFrEF and with or without type 2 diabetes (EMPEROR-Reduced) [86]. These effects were consistent regardless of the presence or absence of diabetes.

Regarding HFpEF, concentrations of BNP, a biomarker of heart failure, decreased after the initiation of treatment with either luseogliflozin, an SGLT2 inhibitor, or voglibose, an alpha-glucosidase inhibitor, in patients with type 2 diabetes mellitus and HFpEF (MUSCAT-HF study) [87]. Recently, it was shown that sotagliflozin reduced the risks of death from cardiovascular causes, hospitalization for heart failure, and an urgent visit for heart failure among patients with diabetes mellitus and a recent worsening of HFrEF or HFpEF [88]. Empagliflozin also significantly reduced the primary composite endpoint of cardiovascular death or hospitalization for worsening heart failure compared to that in the placebo group in patients with HFpEF and with or without type 2 diabetes (EMPEROR-Preserved) [89]. With regard to the pathophysiological mechanism underlying the beneficial effects of SGLT2i, the diuretic action of SGLT2i deserves some consideration, since the results of EMPEROR-Preserved may support such a contributory role [90]. The 21% lower risk in the composite primary outcome obtained with empagliflozin was mainly driven by a reduction in HF hospitalizations, since no significant difference in cardiovascular death was recorded.

SGLT2 inhibitors reduced sudden cardiac death in the EMPA-REG OUTCOME trial [69]. There are several possible mechanisms for the reduction in sudden cardiac death. One possible mechanism is an increase in serum magnesium concentration due to SGLT2 inhibition [91]. Hypomagnesemia tends to cause premature ventricular contractions, particularly in diabetic patients [92]. Serum Mg increased by 0.1 mEq/L (0.05 mmol/L) during empagliflozin treatment in the EMPA-REG OUTCOME study. These effects might contribute to the prevention of sudden cardiac death. As another possible mechanism, SGLT’s downregulation by SGLT2 inhibitors has been reported [93]. In a study using iPS-derived cardiomyocytes, high-glucose culture increased the cell size, SGLT1/SGLT2 was upregulated, and Ca^2+^ flowed into the cells via the sodium–calcium exchanger. Intracellular Ca overload causes delayed afterdepolarizations (DADs) and is prone to arrhythmia, but empagliflozin treatment downregulates SGLT1/SGLT2 and these effects are canceled. Another possible mechanism is the attenuation of late-*I*_Na_ by SGLT2 inhibitors, as mentioned above [81]. The induction of late-*I*_Na_ contributes to action potential prolongation, calcium loading, and the generation of early and delayed afterdepolarizations. Therefore, the inhibition of late-*I*_Na_ by SGLT2 inhibitors may also contribute to the prevention of sudden cardiac death.

### 11.8. Glucagon-like Peptide-1 (GLP-1) Receptor Agonists

SGLT-2 inhibitors and GLP-1 receptor agonists lowered all-cause mortality, cardiovascular mortality, non-fatal myocardial infarction, and kidney failure, but GLP-1 receptor agonists had little or no effect on hospitalizations for heart failure [94].

## 12. Conclusions

The pathophysiology of diabetic cardiomyopathy and treatments for heart failure with diabetes reported so far are presented in this review. Treatment with an SGLT2 inhibitor is currently one of the most effective treatments for heart failure associated with diabetes. However, an effective treatment for the lipotoxicity of the myocardium has not been established, and the establishment of an effective treatment is needed in the future.

## Figures and Tables

**Figure 1 ijms-23-03587-f001:**
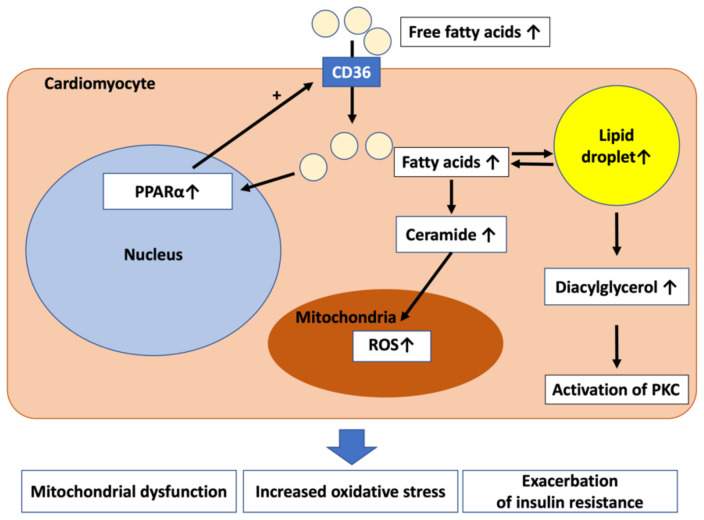
Lipotoxicity to the myocardium in diabetic cardiomyopathy. CD36, cluster of differentiation 36; PKC, protein kinase C; PPARα, peroxisome proliferator-activated receptor α; ROS, reactive oxygen species.

**Figure 2 ijms-23-03587-f002:**
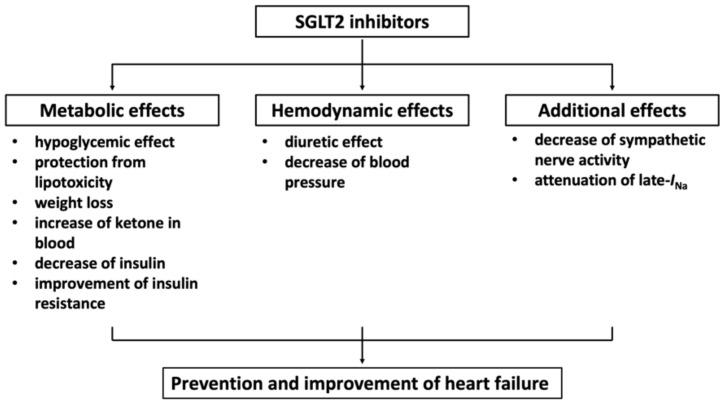
Mechanisms by which SGLT2 inhibitors prevent and improve heart failure. SGLT2, sodium glucose cotransporter 2; late-*I*_Na_, late component of the cardiac sodium channel current.

**Table 1 ijms-23-03587-t001:** Pathogenesis of heart failure associated with diabetes.

1. Coronary artery disease
2. Ischemia due to capillary disorders (abnormal microcoronary circulation)
3. Increased myocardial fibrosis and myocardial hypertrophy
4. Increased activity of the renin–angiotensin–aldosterone system (RAAS)
5. Impaired myocardial energy metabolism and lipotoxicity
a. Decrease in myocardial glucose utilization due to absolute and relative insulin deficiency
b. Increased uptake of fatty acids, increased intermediate products and lipotoxicity
6. Increased oxidative stress due to advanced glycation end products (AGEs), increased activity of RAAS and mitochondrial dysfunction
7. Mitochondrial dysfunction
8. Inflammation
9. Abnormal myocardial calcium handling
10. Autonomic dysregulation in the heart
11. Sodium retention due to hyperinsulinemia

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
