# Peer review of "Pathophysiology and Treatment of Diabetic Cardiomyopathy and Heart Failure in Patients with Diabetes Mellitus"

_ijms, 2022, doi:10.3390/ijms23073587_

Round 1

Reviewer 1 Report

This review is entitled 'Pathophysiology and Treatment of Diabetic Cardiomyopathy'. The topic is of significant interest from a clinical and scientific point of view.

1. Diabetic cardiomyopathy was first described by Rubler in 1972. His description meets the contemporary definition of cardiomyopathy of the European Society of Cardiology. The ESC is not very consistent in reference 2.
2. Reference 11 is often cited in relation to the history of diabetic cardiomyopathy. However, when one reads this 1954 Lancet  paper, no convincing case can be made that this manuscript describes an entity related to the contemporary definition of diabetic cardiomyopathy.
3. Diabetic cardiomyopathy can also present as HFrEF. Please define HFrEF and HFpEF.
4. The effect of empafliglozin in EMPEROR-Preserved was, contrary to what is generally accepted, not very convincing. After all, the effect of the composite endpoint was mainly driven by reduced hospitalizations for worsening heart failure.
5. The authors describe the effects of several drugs. However, diabetic cardiomyopathy is a specific entity whereas the drugs described may or may not have effects on different cardiovascular endpoints in subjects with diabetes. That is a different question. 
6. BNP is only a biomarker. 
7. In general, I doubt whether this review is specific for diabetic cardiomyopathy. It is rather a review on cardiovascular disease in subjects with diabetes mellitus.

Author Response

We would like to resubmit our manuscript entitled "Pathophysiology and Treatment of Diabetic Cardiomyopathy and Heart Failure in Patients with Diabetes Mellitus " to Molecular Pathways in Cardio–Metabolic Disease", International Journal of Molecular Sciences.

The editor and the reviewer kindly pointed out several parts in our previous manuscript that should be clarified, and we have revised these parts.

The answers to the reviewer’s comments are as follows.

Reviewer #1:

We greatly appreciate the reviewer’s comments.

Comment 1: Diabetic cardiomyopathy was first described by Rubler in 1972. His description meets the contemporary definition of cardiomyopathy of the European Society of Cardiology. The ESC is not very consistent in reference 2.

Answer: In accordance with the reviewer’s comment, we added the following sentence: “His description meets the contemporary definition of cardiomyopathy of the European Society of Cardiology [13].” (page 2, 2nd paragraph).

Comment 2: Reference 11 is often cited in relation to the history of diabetic cardiomyopathy. However, when one reads this 1954 Lancet paper, no convincing case can be made that this manuscript describes an entity related to the contemporary definition of diabetic cardiomyopathy.

Answer: In accordance with the reviewer’s comment, we deleted the sentence: “In 1954, Lundbaek suggested that diabetic patients may develop heart disease without developing coronary artery occlusion [11].” (page 2, 2nd paragraph).

Comment 3: Diabetic cardiomyopathy can also present as HFrEF. Please define HFrEF and HFpEF.

Answer: In accordance with the reviewer’s comment, we added the following sentence: “and left ventricular systolic dysfunction, heart failure with reduced ejection fraction (HFrEF) often appears as diabetes progresses.”(page 2, 3rd paragraph).

Comment 4: The effect of empafliglozin in EMPEROR-Preserved was, contrary to what is generally accepted, not very convincing. After all, the effect of the composite endpoint was mainly driven by reduced hospitalizations for worsening heart failure.

Answer: In accordance with the reviewer’s comment, we added the following sentences: “With regard to the pathophysiological mechanism underlying the beneficial effects of SGLT2i, diuretic action of SGLT2i deserves some consideration, since the results of EMPEROR-Preserved may support such a contributory role [85]. The 21% lower risk in the composite primary outcome obtained with empagliflozin was mainly driven by a reduction in HF hospitalizations, since no significant difference in cardiovascular death was recorded.”(page 10, 1st paragraph).

Comment 5: The authors describe the effects of several drugs. However, diabetic cardiomyopathy is a specific entity whereas the drugs described may or may not have effects on different cardiovascular endpoints in subjects with diabetes. That is a different question. 

Answer: In accordance with the reviewer’s comment, we added the following sentences:  “We describe the effects of several drugs in this review. However, diabetic cardiomyopathy is a specific entity, while the drugs described below may or may not have effects on different cardiovascular endpoints in subjects with diabetes.” (page 6, 3rd paragraph).

Comment 6: BNP is only a biomarker. 

Answer: In accordance with the reviewer’s comment, we added the following words: “concentrations of BNP, a biomarker of heart failure,” (page 9, last paragraph).

Comment 7: In general, I doubt whether this review is specific for diabetic cardiomyopathy. It is rather a review on cardiovascular disease in subjects with diabetes mellitus.

Answer: In accordance with the reviewer’s comment, we change the title to “Pathophysiology and Treatment of Diabetic Cardiomyopathy and Heart Failure in Patients with Diabetes Mellitus”.

We hope that these changes are sufficient to satisfy the editor and the reviewers and that this manuscript will now be acceptable for publication in International Journal of Molecular Sciences.

Thank you for your consideration of our paper. 

Reviewer 2 Report

Although diabetic cardiomyopathy (DCM) is a serious problem in diabetes, the definition and characteristics are vague and still controversial. That limits the understanding of the underlying mechanisms or the development of effective treatment. It is important to overview and to discuss about the possible mechanisms and potential strategies based on the treatments currently available.

This manuscript reviews pathophysiology of diabetic heart disease based on the knowledge mainly in the clinical settings. One major concern is that it seems to be a little superficial for the review of diabetic cardiomyopathy. If this is indeed focusing on the diabetic cardiomyopathy, a precise reconstruction of the outline and an extensive revision would be essential to make the perspective clear. 

Major Concerns:
This could be a good introductory article overviewing diabetic heart disease for clinical practice. If the authors target such readers, the title and the abstract are misleading. Please make it clear the points who would be the expected readers and what is the aim of this article.

Specific comments:
1 Please add a few sentences clarifying the aims and the scope of this review article in the abstract.

2 Based on the aims, please reconsider the outline to deliver the precise arguments. Please make sure those arguments will be supported with multiple sources.

3 Does Table 1 really help to summarize the topic?
May want to list only the pathogenesis for diabetic cardiomyopathy. I would also suggest reformatting and making it easier to count how many bullet points listed.

4 The paragraph explaining about Fig1 (line 94-) is too brief. Please add some phrases or sentences to elaborate the items, such as DG, ceramide, sphingolipids.

5 Section 11 mostly discusses about vascular endothelial dysfunction. How is this related to DCM which should be an independent event from coronary artery impairment?

6 There is a really nice review article for the Diabetic cardiomyopathy(1). It looks not cited.

Reference
1.      Ritchie RH, Abel ED. Basic Mechanisms of Diabetic Heart Disease. Circ Res. 126(11):1501-1525. doi:10.1161/CIRCRESAHA.120.315913

Author Response

We would like to resubmit our manuscript entitled "Pathophysiology and Treatment of Diabetic Cardiomyopathy and Heart Failure in Patients with Diabetes Mellitus " to Molecular Pathways in Cardio–Metabolic Disease", International Journal of Molecular Sciences.

The editor and the reviewer kindly pointed out several parts in our previous manuscript that should be clarified, and we have revised these parts.

The answers to the reviewer’s comments are as follows.

Reviewer #2:

We greatly appreciate the reviewer’s comments.

Comment: Major Concerns: This could be a good introductory article overviewing diabetic heart disease for clinical practice. If the authors target such readers, the title and the abstract are misleading. Please make it clear the points who would be the expected readers and what is the aim of this article.

Answer: In accordance with the reviewer’s comment, we change the title to “Pathophysiology and Treatment of Diabetic Cardiomyopathy and Heart Failure in Patients with Diabetes Mellitus”.

Specific comments:
Comment 1: Please add a few sentences clarifying the aims and the scope of this review article in the abstract.

Answer: In accordance with the reviewer’s comment, we added the following sentence: “This review provides an overview of heart failure in diabetic patients for clinical practice to clinicians”(abstract and page 2 1st paragraph).

Comment 2: Based on the aims, please reconsider the outline to deliver the precise arguments. Please make sure those arguments will be supported with multiple sources.

Answer: In accordance with the reviewer’s comment, we added the following sentences:  “We describe the effects of several drugs in this review. However, diabetic cardiomyopathy is a specific entity, while the drugs described below may or may not have effects on different cardiovascular endpoints in subjects with diabetes.” (page 6, 3rd paragraph).

Comment 3: Does Table 1 really help to summarize the topic?
May want to list only the pathogenesis for diabetic cardiomyopathy. I would also suggest reformatting and making it easier to count how many bullet points listed.
Answer: In accordance with the reviewer’s comment, we added the following sentence: “Table 1 shows the currently considered mechanism of heart failure associated with diabetes and it is not limited to diabetic cardiomyopathy. (page 6, 3rd paragraph)”  and numbered each item in Table 1. 

Comment 4: The paragraph explaining about Fig1 (line 94-) is too brief. Please add some phrases or sentences to elaborate the items, such as DG, ceramide, sphingolipids.
Answer: In accordance with the reviewer’s comment, we added the following sentences: “Diacylglycerol causes exacerbation of insulin resistance and oxidative stress through activation of protein kinase C (PKC). The level of diacylglycerol is increased, accompanied by increased membrane localization of PKC and decreased Akt activity in human failing myocardium [26]. These observations suggest that diacylglycerol is a toxic lipid intermediate in the heart. Ceramide causes mitochondrial dysfunction and oxidative stress. C6-ceramide reduces Akt activity and increases brain natriuretic peptide (BNP) mRNA expression in cardiomyocytes, and inhibition of ceramide biosynthesis in the heart improves lipotoxic cardiomyopathy [27]. These results suggest that ceramide accumulation contributes to the development of left ventricular hypertrophy and cardiac dysfunction.” (page 3, last paragraph).

Comment 5: Section 11 mostly discusses about vascular endothelial dysfunction. How is this related to DCM which should be an independent event from coronary artery impairment?
Answer: In accordance with the reviewer’s comment, we added the following sentence: “Endothelial dysfunction is observed in patients with heart failure [42] and it is associated with an increased mortality risk in subjects with both ischemic and nonischemic heart failure [43].“ (page 5, last paragraph).

Comment 6: There is a really nice review article for the Diabetic cardiomyopathy(1). It looks not cited.
Reference
1.      Ritchie RH, Abel ED. Basic Mechanisms of Diabetic Heart Disease. Circ Res. 126(11):1501-1525. doi:10.1161/CIRCRESAHA.120.315913

Answer: In accordance with the reviewer’s comment, we cited the article (reference #23).

We hope that these changes are sufficient to satisfy the editor and the reviewers and that this manuscript will now be acceptable for publication in International Journal of Molecular Sciences.

Thank you for your consideration of our paper.

Reviewer 3 Report

It was a pleasure to read this very good written and well-organized review. I do not have much to add since I feel this work might be of great interest for the scientific community.

Minor comment -

There are few instances where correction is needed for gene /protein nomenclature.

Author Response

We would like to resubmit our manuscript entitled "Pathophysiology and Treatment of Diabetic Cardiomyopathy and Heart Failure in Patients with Diabetes Mellitus " to Molecular Pathways in Cardio–Metabolic Disease", International Journal of Molecular Sciences.

The editor and the reviewer kindly pointed out several parts in our previous manuscript that should be clarified, and we have revised these parts.

The answers to the reviewer’s comments are as follows.

Reviewer #3:

We greatly appreciate the reviewer’s comments.

Minor comment -There are few instances where correction is needed for gene /protein nomenclature.

Answer: In accordance with the reviewer’s comment, we added the following abbreviations:

AMPK             AMP-activated protein kinase

BNP                brain natriuretic peptide

IL-1β               interleukin-1 beta

IL-18               interleukin-18

NADPH          nicotinamide adenine dinucleotide phosphate

We hope that these changes are sufficient to satisfy the editor and the reviewers and that this manuscript will now be acceptable for publication in International Journal of Molecular Sciences.

Thank you for your consideration of our paper. 

Round 2

Reviewer 1 Report

The authors have done a satisfactory effort to address the concerns of the reviewer.

Reviewer 2 Report

All the major concerns have been addressed.